# Toxic Algal Bloom Recurrence in the Era of Global Change: Lessons from the Chilean Patagonian Fjords

**DOI:** 10.3390/microorganisms11081874

**Published:** 2023-07-25

**Authors:** Patricio A. Díaz, Rosa I. Figueroa

**Affiliations:** 1Centro i~mar, Universidad de Los Lagos, Casilla 557, Puerto Montt 5480000, Chile; 2Centre for Biotechnology and Bioengineering (CeBiB), Universidad de Los Lagos, Casilla 557, Puerto Montt 5480000, Chile; 3Centro Oceanográfico de Vigo, Instituto Español de Oceanografía (IEO-CSIC), Subida a Radio Faro 50, 36390 Vigo, Spain; rosa.figueroa@ieo.csic.es

**Keywords:** harmful algal blooms, climate change, phycotoxins, fish-killing, socioeconomic impacts, Chilean Patagonia

## Abstract

Toxic and harmful algal blooms (HABs) are a global problem affecting human health, marine ecosystems, and coastal economies, the latter through their impact on aquaculture, fisheries, and tourism. As our knowledge and the techniques to study HABs advance, so do international monitoring efforts, which have led to a large increase in the total number of reported cases. However, in addition to increased detections, environmental factors associated with global change, mainly high nutrient levels and warming temperatures, are responsible for the increased occurrence, persistence, and geographical expansion of HABs. The Chilean Patagonian fjords provide an “open-air laboratory” for the study of climate change, including its impact on the blooms of several toxic microalgal species, which, in recent years, have undergone increases in their geographical range as well as their virulence and recurrence (the species *Alexandrium catenella*, *Pseudochattonella verruculosa*, and *Heterosigma akashiwo*, and others of the genera *Dinophysis* and *Pseudo-nitzschia*). Here, we review the evolution of HABs in the Chilean Patagonian fjords, with a focus on the established connections between key features of HABs (expansion, recurrence, and persistence) and their interaction with current and predicted global climate-change-related factors. We conclude that large-scale climatic anomalies such as the lack of rain and heat waves, events intensified by climate change, promote the massive proliferation of these species by creating ideal conditions for their growth and persistence, as they affect water-column stratification, nutrient inputs, and reproductive rates.

## 1. Introduction

Harmful algal blooms (HABs) have become a major problem in coastal areas worldwide. As the aquaculture industry assumes a greater role in global food security and nutrition, the rising number of HABs and their adverse impacts, responsible for bans on shellfish harvest, massive fish kills, and overall ecosystem deterioration, have stimulated research into the culprit factors underlying these events. While the role of climate change has been considered, a direct connection between climate change and HABs has been established only on a local scale [1].

Chile is highly vulnerable to the impacts of climate change. By 2040, it is predicted to be the most water-stressed country in the Western Hemisphere [2]. During the past decade, Chile has been exposed to droughts, wildfires, a glacial retreat, and extreme weather conditions. However, the rising number and magnitude of HABs that have occurred during this period should also be included in this list, given their damage to the environment and the regional economy. Among the bloom-forming species of concern is toxin-producing *Alexandrium catenella*, responsible for paralytic shellfish poisoning (PSP), which has been geographically expanding from south to north, blooming in previously unaffected areas. Moreover, this expansion is being consolidated in the sediments of these new areas, where resting stages of this species have been found, guaranteeing a “seed bank” for potential new blooms [3,4,5,6]. Changing climatic conditions and increasing water temperatures are expected to promote the expansion of *A. catenella* [7,8] and of other ichthyotoxic species, such as *Heterosigma akashiwo* and *Pseudochattonella verruculosa*, outbreaks of which have also occurred under conditions associated with climate change [9,10,11]. A bloom of *P. verruculosa* in 2016 affected 45 salmon farms, causing losses of around USD 800 million [12,13]. During March and April 2021, a microalgal bloom of *H. akashiwo* in the Comau Fjord, NW Patagonia, affected ten salmon farms and led to the loss of 6000 tons of salmon [11,14].

The vast majority of HAB events, including the most intense ones, have occurred in the region of the fjords and channels of Chilean Patagonia, which includes the three southernmost administrative regions of Chile, Los Lagos, Aysén, and Magallanes (Figure 1). This extensive area, one of the world’s most extensive fjord and channel systems, is characterized by an abrupt bathymetry and a complex coastal morphology that are both strongly shaped by oceanic water [15]. A marked water-column stratification is determined by freshwater input and by seasonal and latitudinal patterns in precipitation, which includes rainfall and ice melting [15,16,17]. The wind regime is dominated, on average, by westerly winds [18,19], although the synoptic scale variability also plays an important role in the weather conditions of the region [20]. High precipitation, from 3500 to 4000 mm yr^−1^, can occur over 200 days per year [21], but can be enhanced up to ~5000 mm yr^−1^ inland due to the orographic uplift of the Austral Andes [22].

Despite the environmental and socioeconomic impacts of these bloom events, the role of climate change in fostering their development remains poorly understood, because it is currently impossible to separate the influence of climate change from other anthropogenic impacts. HABs are caused by a combination of multiple factors, but the positive effect of rising temperatures on the microalgal growth rate, leading to earlier and longer-lasting blooms [7], is relevant only after other variables, such as water-column mixing and nutrient inputs, have been allowed to act. Under laboratory conditions, a combination of temperature and salinity is the key to the maximal growth of *H. akashiwo* and *P. verruculosa* [23], but the main environmental factor responsible for the intensity and duration of these blooms has been shown to be climate anomalies. During 2016 and 2021, an extremely sunny and dry summer—caused by El Niño events and positive phases of the Southern Annular Mode (SAM)—resulted in a strong uplift, water-column mixing, and nutrient inputs into the fjords, fostering blooms of these fish-killing species. Detailed analyses of the mechanisms and environmental/anthropogenic triggers that generate HABs in the Chilean Patagonia can therefore be useful in drawing lessons regarding future HAB events in this and other vulnerable regions of the world.

## 2. Climate Modulation: A Global Phenomenon Affecting Microalgal Local Responses

The Patagonian fjords are subject to multidecadal fluctuations related to large-scale climate cycles, such as the El Niño Southern Oscillation (ENSO), the Pacific Decadal Oscillation (PDO), and the SAM [24,25,26,27]. The ENSO is a recurring climate pattern across the tropical Pacific with warm (El Niño) and cool (La Niña) phases. Thus, the equatorial Pacific Ocean experiences a periodic fluctuation in sea surface temperature and air pressure. Likewise, the PDO is sometimes referred to be a persistent pattern of Pacific climatic fluctuation that resembles El Niño, with important surface climate anomalies over the mid-latitude South Pacific Ocean, including South America [28,29]. The SAM is the dominant mode of extratropical climate variability in the Southern Hemisphere, linked to fluctuations in the strength and location of the polar jet around Antarctica [30]. The positive SAM trend in summer observed in recent decades explains much of the contemporaneous drying observed in Northern Patagonia [25,31]. These cycles (Figure 2) affect regional systems directly, through changes in local wind patterns, rainfall, and probably through changes in the velocity of glacier-melting, one of the main sources of freshwater inputs (and water-column stability). The modification of local atmospheric processes—the main hydrodynamic physical forcing in the Patagonian fjords and other coastal areas—has direct effects on the seasonal phytoplankton succession, because the latter is highly responsive to the stratification and mixing of the water column [17,32,33]. Thus, the microscale physical–biological interactions in fjords and semi-enclosed systems are modulated both temporally and spatially by processes such as turbulence, tidal cycles, and circadian rhythms [34,35].

The potential impacts of climate variability on the multi-decadal fluctuations in the composition and abundance of different functional groups, and on their respective trophic levels, have been the focus of attention in recent decades, due to the threat of climate change and global warming [36]. The effects will be exacerbated in highly heterogeneous systems such as the Patagonian fjords, where multiple niches promote the development or aggregation of different species, including microalgae [37,38]. Long-term variations in some phytoplanktonic species have been associated with anomalous oceanographic events that occur periodically over decadal time scales [39]. Different HAB species, even those belonging to the same genus, may exhibit distinct responses to changing environmental conditions [40], which has complicated predictions on the impact of climate change on different species. Hallegraeff [41] hypothesized that, under a global climate change scenario, some HAB species will become more competitive, whereas others will decline with restrictions in their distribution area.

During the last 50 years, positive phases of the ENSO index during summer have generated the optimal environmental conditions for the occurrence of intense microalgal blooms in Chilean Patagonia. Examples include blooms of the dinoflagellate *A. catenella* during the summers of 1996, 1998, 2002, 2006, 2009, and 2016 in the fjords and channels of NW Patagonia [6,12,42,43]. The El Niño of 2015/16 was one of the most intense to occur in recent decades (Figure 2) and was comparable to the events of 1982/83 and 1997/98. The warming of the equatorial Pacific Ocean impacted weather patterns around the world (Figure 3) and generated the perfect climate scenario for the occurrence of one of the most intense HAB events in Chilean Patagonia, with severe socioeconomic impacts that extended worldwide [9,13,44,45]. In NW Patagonia, January 2016 had the highest solar radiation and lowest wind intensity of the last 70 years, according to Garreaud [46]. The large-scale climate anomalies, mostly forced by strong El Niño-related atmospheric teleconnections superimposed on the positive phase of the SAM created optimal conditions for the formation of intense algal blooms (e.g., *P. verruculosa* and *A. catenella*), including weakened wind strength, negative rainfall anomalies [46], and a high positive sea surface temperature (SST), with positive anomaly values of up to 3 °C (Figure 4). The local response to this global modulation also included extreme drought, characterized by a reduction in precipitation (50%), and thus a significant reduction in river discharge into coastal waters. The combined reduction in freshwater inputs into the coastal system weakened the superficial haline stratification, allowing the entry of deeper water of higher salinity and rich in nutrients into superficial layers of the inland sea of NW Patagonia, exacerbated by the continuous upwelling pulses of oceanic water masses that occurred on the adjacent continental shelf. A laboratory simulation of these high salinity and temperature conditions showed that they are optimal for the growth of *P. verruculosa,* with optimal growth conditions at 18 °C and salinity of 30 psu [23,47].

## 3. Extreme HAB Events in Chilean Patagonia

Chile is the world’s second-largest producer of cultured blue mussels (*Mytilus chilensis*) and salmon (mainly *Salmo salar* and *Oncorhynchus kisutch*), with 400,000 and 1,000,000 tons per year, respectively. In the following sections, several of the high-intensity HABs that have occurred in Chilean Patagonia in recent decades are discussed, including those responsible for biotoxin-mediated fish kills (*Pseudo-nitzchia*,* Dinophysis,* and *Alexandrium* genus), fish-killing species (*Heterosigma akashiwo, Pseudochattonella verruculosa,* and *Karenia *sp.), and those non-toxic but causing harm to the fish because they reach high-biomass microalgal accumulations (*Lepidodinium chlorophorum* and *Prorocentrum micans*). The relationship of these events to environmental and anthropogenic factors and the species-specific responses to them are considered as well.

### 3.1. HABs of Biotoxin-Producing Microalgae

Toxin-producing HAB species represent the main natural threat to bivalve farms and public health. The toxins are accumulated by filter-feeding bivalves and may accumulate to levels that make the shellfish unfit for human consumption (regulatory level), such that bans on their extraction are imposed by health and fishery authorities. Along the coasts of the fjords and channels of Chilean Patagonia, proliferations of diatoms of the genus *Pseudo-nitzschia*, responsible for amnesic bivalve poisoning (ASP), and of dinoflagellates of the genera *Dinophysis* and *Alexandrium*, agents of diarrheic shellfish poisoning (DSP) and PSP, respectively, are recurrent and cause considerable impacts on human health, as well as losses to artisanal fishing and to the powerful mussel industry [12,48].

In recent years, a northwards expansion of HAB events, in particular of PSP outbreaks caused by the dinoflagellate *A. catenella*, has been observed in southern Chile, being particularly important the blooms of the years 2006, 2009, and 2016, in which cell densities up to 6 × 10^6^ cells L^−1^ were reached [6,42,43,49]. Both meteorological conditions [46] and nutrient inputs from aquaculture practices [50] have been implicated in these outbreaks, but definitive evidence is lacking. During the summer–autumn of 2016, toxic outbreaks reached as far north as Los Ríos (39° S), the first such occurrence [44,51]. PSP outbreaks pose a major threat to public health and to the fishing industry in the Patagonian fjords, in particular in the Aysén and Magallanes regions [52]. Outbreaks in the Los Lagos region were recorded in late summer–early autumn of 2002 [42], 2009 [49], 2016 [51], and 2018 [53] and were attributed to a northward expansion of those from Aysén. The 2018 event in the Aysén region was marked by the highest toxicity in mussels (*Aulacomya atra*) reported worldwide, 143,000 μg STX eq. 100 g^−1^. Recently, toxic cells of *A. catenella* were detected in the Bío-Bío region (39° S), confirming the northward expansion that has occurred from the Magallanes region (56° S) over the last five decades [54]. The Bío-Bío region is the second most important area for the artisanal fishing of benthic resources in Chile [55]. While high concentrations of PSP toxins in shellfish have not yet been reported in the region, should they occur, their socioeconomic impact will be severe. In fact, recent intense episodes of lipophilic [56] and amnesic toxins in this region (years 2019 and 2022, respectively) led, in the latter case, to closures of shellfish extraction.

In Southern Chile, the main threat to aquaculture is posed by endemic species of the genus *Dinophysis* (*D. acuta*, *D. acuminata* complex), producers of the lipophilic toxins okadaic acid, OA, and derivatives, pectenotoxins, PTX [48,57], that are the causative agents of DSP. Lipophilic toxin outbreaks in Chile have been especially relevant in Los Lagos, Aysén, and Magallanes since the 1990s, although the first recorded event was in 1970, with gastrointestinal disorders developing in over 100 people after eating ribbed mussels (*A. atra*) from Reloncaví Sound (Los Lagos region). To date, the toxins OA, DTX1, and PTX2 have been detected in shellfish (mainly mytilids) from the Aysén region [58,59,60], but single-cell toxin analysis has yet to demonstrate the relative contribution of each species. However, it was in this region, specifically at Puyuhuapi Fjord, where, during the summer of 2018, the most intense bloom of *D. acuta*, in terms of cell density, was recorded worldwide [61,62].

Diatoms of the genus *Pseudo-nitzschia*, mainly *P. australis*, produce the toxin ASP, but toxic events are generally of short duration, and toxin levels that exceed the permissible limits for human consumption in shellfish [12,63] are rarely reached. Nevertheless, in the summer of 2020/21 (December to March), an intense bloom of *P. australis* and *P. pseudodelicatissima* caused prolonged sanitary bans in the Los Lagos region, impacting artisanal fishing and bivalve aquaculture. During the ban, the measured concentration of the toxin domoic acid in shellfish was as high as 140 mg kg^−1^ [64].

The above-described HABs events, despite their intensity and substantial socioeconomic impacts, were monospecific. However, in the summer of 2022, a multi-specific toxic bloom was recorded in Quitralco Fjord, NW Patagonia [65]. This bloom was generated by four toxic species, *A. catenella* (7 × 10^3^ cells L^−1^), *D. acuminata* (6 × 10^3^ cells L^−1^), *Protoceratium reticulatum* (18 × 10^3^ cells L^−1^), and *Pseudo-nitzschia* cf. *australis* (2 × 10^6^ cells L^−1^), and resulted in intense brown patches in waters close to the head of the fjord, as well as the death of juvenile hake. Given the niche differences of these HAB species [37,38], while the event was highly anomalous, the potential for its recurrence, and the consequences thereof, cannot be ignored.

### 3.2. Fish-Killing Microalgal Blooms

The proliferation of microalgae that results in massive mortalities in farmed fish, such as *P. verruculosa*, *H. akashiwo*, and *Karenia* spp., although sporadic, poses the greatest danger to the Chilean salmon farming industry [9,11,13], the second-largest farmed salmon producer in the world [55]. Blooms of species such as *Letocylimdrum minimus* and *A. catenella* have repeatedly affected salmon farming in different areas, but none achieved the mortality and socioeconomic impact of the exceptional bloom of the noxious species *P. verruculosa* in the summer of 2016 [9,66]. During that event, the harmful effects of this species caused the loss of 39,942 tons of salmon [13,67]. In addition to the significant economic consequences, with losses of USD 800 million [12], the outbreak demonstrated the vulnerability of the salmon farming industry to such large-scale natural events and the urgent need to implement measures to counteract them. Among the measures subsequently implemented in Chile, with the aim of reducing future risks, were the establishment of contingency plans for the management of massive fish mortalities [68], changes in the design and/or monitoring tools, and the evaluation of prevention, control, and mitigation measures.

One of the first blooms to cause high mortality of Chilean salmon was that of *H. akashiwo*, which, in September 1988, affected farming centers in the Chiloé Inland Sea and Reloncaví Sound [69]. This event caused the loss of 5000 tons of salmon, corresponding to 50% of the annual production. *H*. *akashiwo* is a cosmopolitan species of raphidophyte that forms blooms in coastal marine and brackish waters, with occasional fish kills associated with the production of reactive oxygen species (ROS) and/or polyunsaturated fatty acids (PUFAs) [70,71]. This species was responsible for an intense surface bloom (max. 200,000 cells mL^−1^) within Comau Fjord, NW Patagonia, that led to the loss of 6000 tons of salmon at the beginning of autumn 2021 [11,14]. This event was modulated by global climatic anomalies, including extremely dry conditions, high solar radiation, and strong southerly winds, coupled with local processes, such as long water-retention times inside the fjord and the local physical uplift process favored by the N-S orientation of the fjord [11]. An outbreak of *H. akashiwo* was also recorded on the west coast of the island of Chiloe at the end of the summer of 2022. The bloom (max. 15,500 cells mL^−1^) formed clearly visible surface patches (Figure 5) and caused massive salmon mortalities.

The high cell densities (10^6^–10^8^ cells L^−1^) reached by species of the genus *Karenia* (e.g., *Karenia mikimotoi*) induce the production of hemolytic exotoxins that can cause mortality in fish and mollusks [72]. Events associated with species of this group in Patagonian fjords were first recorded in the early 2000s [73], but, in the last decade, they have become increasingly recurrent. In the summer of 2017, *Karenia* species were responsible for intense salmon mortalities in the Gulf of Penas (Aysén region), after cells of these dinoflagellates were transported in well-boats traveling to and from the Magallanes region. Villanueva et al. [74] estimated that a January 2017 event was responsible for the mass mortality of around 170,000 salmon, worth USD 390,000. In the summers of 2020 and 2023, blooms were detected within fjords of the Aysen region, such as Pitipalena Fjord and Puyuhuapi Fjord (unpublished data). Considering the growth conditions preferred by *Karenia* species, with a high growth rate achieved at a salinity of 30 and a temperature of 18 °C [75,76], the compatible environment of the fjord system may increase the risk of outbreaks. These environmental conditions are not typical of the fjord system, especially in areas with high contributions of fresh water from mighty rivers where there is strong haline stratification in the first 10 m, such as the Reloncaví Fjord in Los Lagos region, but they can occur during periods of extreme hydroclimatic anomalies such as those that occurred in the summer of 2016 [46].

### 3.3. High-Biomass Blooms

High biomass HABs (HB-HABs) are defined as those caused by “non-toxin-producing species but whose accumulated biomass can be toxic”. Despite their intensity, these have been scarcely studied in Chile [77,78], as their impacts on public health and aquaculture have been much smaller than those resulting from toxin-producing and fish-killing blooms. However, blooms of the dinoflagellates *Lepidodinium chlorophorum* and *Prorocentrum micans*, which result in “green tides” and “brown tides”, respectively, have been recorded with increasing frequency in Reloncaví Sound and the Chiloé Inland Sea (years 2020 and 2022, respectively). A study of the environmental variables that trigger HB-HABs identified an increase in the concentration of nutrients in the water column, mainly ammonium, from the degradation of organic matter [79].

Data on economic losses due to the mortality of farmed fish following HB-HABs are likewise scarce. In March 2021, high Atlantic salmon (*Salmo salar*) mortalities (~162,000 individuals, ~80 tons) were recorded in the area of the Chiloé Inland Sea (Apiao, Quinchao, and Reñihue Fjords), associated with an intense bloom of *L. chlorophorum*. In Reñihue Fjord alone, economic losses were close to USD 3.5 million. The previous year, an intense bloom of the same species covered the entire inland sea of the Aysén and Los Lagos regions (Figure 6), with cell densities inside Reloncaví Sound reaching ~6 × 10^6^ cells L^−1^ towards the end of summer and beginning of autumn 2020 [80]. Similarly, in February 2022, an intense bloom of *P. micans* was detected inside the Reloncaví Sound and Ancud Gulf, generating intense brown surface patches due to cell densities that exceeded 3.4 × 10^6^ cells L^−1^. Although no mortalities of farmed fish were reported, behavioral changes (e.g., lack of appetite) were observed. A public alarm was generated by the discoloration of the water in coastal areas of Reloncaví Sound, including the city of Puerto Montt, Los Lagos region. Noxious odors, resulting from the decomposing organic matter, were reported by residents of the area during the event and sparked the attention of the local community.

Both the behavior and biology of microalgal species in inland waters of southern Chile are poorly understood. Also unclear is the cell density at which negative impacts on the regional aquaculture industry and risks to public health are incurred, which has made it difficult to implement timely management measures aimed at mitigation.

## 4. Learning to Live with HAB: Some Lessons

In the last two decades, important advances have been made in understanding the biology and toxicology of bloom-forming species and in defining the conditions that favor HABs. Yet, important knowledge gaps remain regarding the factors that determine the inter-annual variability of each species, the origin of the inoculum seeding the populations, and the fate of declining, non-cyst-forming populations.

The priority objectives of research programs dedicated to the study of HABs are the unequivocal characterization of toxic species, the identification and parameterization of the different factors that determine where and when HAB species proliferate, and, ultimately, the development of capabilities to predict bloom events. The international GEOHAB (Marine Geological and Biological Habitat Mapping) program, sponsored by the SCOR (Scientific Committee for Ocean Research) and IOC (Intergovernmental Oceanographic Commission of UNESCO), has promoted a comparative ecosystem approach to the study of HABs. As part of that effort, four systems with characteristic hydrodynamic and chemical forces, some shared and some unique, that give rise to HABs were distinguished: (1) upwelling systems; (2) eutrophic systems; (3) fjords and secluded coastal areas; and (4) stratified systems. One of the objectives of the GEOHAB program is to determine whether there is a characteristic or indicator species for each of these systems, or whether the same species is able to develop diverse adaptive strategies that allow it to proliferate in more than one system [34,35,81,82,83,84].

NW Patagonia provides a natural laboratory where the coastal systems recognized globally by the GEOHAB classification can be studied. The Patagonian fjord system can be included within systems 3 and 4. However, the high anthropogenic pressure from aquaculture activities (salmon and mussel farming) and the population increase in coastal areas could generate conditions that make certain areas eutrophic (system 2). In this sense, it is important to clarify that salmon was a non-indigenous species in Chile when the farms started in the area, and, therefore, the possible effects derived from salmon cultivation can be considered an anthropogenic effect, as it is eutrophication. Although some areas are at higher risk of eutrophication than others, e.g., due to high aquaculture loads [85], a recent study showed that the Patagonian fjords have not undergone significant changes in the nutrient concentrations of their water columns in the last 50 years [86], consistent with some features of system 2. Finally, it has been recently recognized that the seasonal upwelling processes on the adjacent oceanic shelf [87,88] give the fjord system conditions consistent with those of system 1.

GEOHAB’s “Core Research Project: HABs in Fjords and Coastal Embayments” identified four key features that contribute to the development of HAB events in coastal zones: (1) the water-residence time in the system; (2) water exchange with adjacent shelf waters; (3) freshwater inputs from rivers, rainfall, and glacier-melting; and (4) pelagic–benthic coupling in shallow areas. All of these features have been studied and parametrized in the Chilean fjord system to explain the occurrence of extreme HAB events in the region in the last decade. Thus, Soto et al. [85] called attention to the eutrophication risk for water bodies housing salmon farms and created risk maps of climate change for the salmon industry in Chilean Patagonia. The highest risk areas were those with the highest residence times, a key feature included by the GEOHAB project. Fjords in those high-risk areas have been the sites of the most intense blooms over the last decade, including blooms of *P. verruculosa* (Reloncaví Fjord), *D. acuta* (Puyuhuapi Fjord), and *H. akashiwo* (Comau Fjord).

The above-discussed bloom of *P. verruculosa*, which caused one of the highest mortalities of farmed fish worldwide (39,942 tons), provided several lessons on how to better deal with extreme HABs. The large biomass of dead salmon that accumulated in a short period of time led to a health alert [66] and, therefore, to the authorized dumping of 4700 tons of rotting salmon in the Pacific Ocean 140 km from the coast. Subsequently, a high-intensity bloom of *A. catenella* was recorded for the first time on the oceanic coast of the Los Lagos region, causing the stranding and massive mortality of various shellfish species [44]. A connection between the two events—the incorporation of nutrients produced by the decomposing salmon and the bloom of *A. catenella*—could not be established with certainty [45,66], but the ensuing crisis on the Chiloé Island paralyzed all fishing, aquaculture, and tourism activities for ~1 month.

Based on the climate projections for the coming years, HABs will become increasingly recurrent, requiring a toolbox of measures designed to minimize their worst impacts. The dumping of rotting fish, a desperate action, is unlikely to be repeated, due to the serious consequences. However, it is appropriate to ask whether we are prepared to deal with similar large-scale HABs and their potential socioeconomic impacts. The *H. akashiwo* event registered at the beginning of autumn 2021, although less severe in terms of mortality (6000 tons) and geographical coverage (Comau Fjord), put the containment protocols and mitigation measures to the test. The new measures included an increase in the storage capacity of dead fish in the farming centers and greater transport and processing capacity for this biomass. In addition, some mitigation systems have been implemented, such as the use of bubble curtains [89]. Given the intensity of the bloom, in which cell densities exceeded 200,000 cells mL^−1^, some aquaculture centers were relocated or their fish were moved to other areas, significantly reducing the impact. As both *H. akashiwo* and *A. catenella* are resting cyst-forming species (discussed below), other measures aimed at reducing the recurrence risk and the formation of hotspot areas could be considered, such as those targeting potential cyst beds in local sediments [90].

## 5. Future Perspectives: Potential Species Responses, Trends, and Unresolved Topics

Wells et al. [91] examined the potential responses of different HAB species, including species present in the coastal zones of Patagonian fjords and channels, to climate change stressors. Sandoval-Sanhueza et al. [23] investigated the physiological responses of the two microalgal species posing the most severe risks to aquaculture (salmon farming) in Southern Chile, *P. verruculosa* [9,13] and *H. akashiwo* [11,14]. Those studies showed the abundant growth of *H. akashiwo* over a wide range of temperatures (12, 15, 18 °C) and salinities (10, 20, 30 psu), and thus the serious threat posed by this species based on its ability to exploit a wide range of ecological niches (sensu [92]). By contrast, *P. verruculosa* was shown to thrive only at high salinity (30 psu) and a narrow temperature range of 15–18 °C, such that it occupies a much more restricted niche, as also reported by Mardones et al. [47]. The latter environmental conditions were met in NW Patagonia in the summer of 2016, associated with intense local hydroclimatic anomalies that were globally modulated by one of the most intense ENSO events in the last five decades [46], and that accounted for the intensity of the resulting massive and highly lethal bloom of *P. verruculosa*.

The conditions determined by Sandoval-Sanhueza et al. [23] to promote the growth of *P. verruculosa* are comparable to those reported by Aguilera-Belmonte et al. [93] for the dinoflagellate *A. catenella*, which achieves maximum growth at 15 °C and 35 psu. Similar environmental conditions, although with salinities closer to 30 psu, are normally found in the islands and channels located to the west of Moraleda Channel, in the Aysén region, the epicenter of *A. catenella* blooms since the mid-1990s [12,42].

In addition to serving as a natural laboratory for the study of coastal systems, NW Patagonia, with its rapidly changing environmental variables and the pressures exerted by aquaculture and other human activities, allows investigations into the effects of climate change at a regional scale, albeit with global implications, including for subpolar and polar zones such as Antarctica. During the last six decades, summer and autumn precipitation in NW Patagonia has sharply declined, by 5–7% per decade [31,94,95]. The dramatic reduction in rainfall in Puerto Montt (Figure 7A), the northern limit of NW Patagonia, has been accompanied by a significant increase in the occurrence of heat waves (Figure 7B), consistent with the increases in the maximum and average air temperatures reached in this area (Figure 7C,D).

Although climate variability is closely related to the occurrence and intensity of HABs in Patagonia [9,10,11,61], the ability to predict bloom events has been hindered by the large deficits in our knowledge concerning the biological interactions that underlie the complex life cycles of the responsible species, and how those interactions are influenced by the changing climate.

Dinoflagellates grow mainly by asexual reproduction and mitosis, but, under certain environmental conditions, they enter a sexual cycle (Figure 8) that allows them to dominate other microalgae, such as diatoms, due to the resulting higher growth rates [96]. A sexual life cycle includes the formation of benthic resting cysts, the germination of which serves as a new inoculum of vegetative cells and promotes new, recombinant genotypes that maximize the likelihood of population success under different environmental conditions. As resting cysts are highly resistant to adverse environmental conditions and remain viable in the sediments for long periods of time, sexual reproduction (genotype variability) that includes resting cyst production (environmental resistance) enables the successful invasion of new regions and habitats. Although very little research has been performed thus far, climate change is expected to impact the sexual cycle of dinoflagellates, including the formation and germination of resting cysts. For example, under laboratory conditions, a combination of high temperature and low salinity increased the rates of zygote formation [97,98] and resting cyst germination [99] in *Alexandrium* and *Gambierdiscus.* While the germination rate is constricted by species-specific temperature windows, it generally increases at the higher end of that range [100]. The sexual encystment of *A*. *catenella* is also enhanced by rising temperatures [101]. In contrast to previous assumptions that sexuality in dinoflagellates was sporadic and unidirectional, recent studies have shown that is in fact both frequent and plastic [102,103], suggesting an essential role of life cycle transitions in the environmental adaptation and competitive success of these organisms.

## 6. Conclusions

The effects of climate variability and climate change on the occurrence, duration, and intensity of HABs are undeniable. In the system of fjords and channels of Chilean Patagonia, HABs are generally associated with positive phases of the main climatic indices (ENSO, SAM). However, on a local scale, global climatic modulation will be further shaped by local factors such as the pressure of anthropic activities, geomorphology, and water-residence times, all of which will influence the development and intensity of HABs. Furthermore, the complex life cycles of bloom-forming species, and the large knowledge gaps regarding the biological interactions and environmental factors that modulate their life cycle transitions, highlight the challenges in understanding the increasing expansion and recurrence of HABs and the effect of climate change. Although organisms can adapt their life cycles to changes in environmental conditions, not all microalgal species will benefit from the expected global changes in climate. However, the growth and subsequent expansion of a species may be favored by compatible climate changes that trigger advantageous life cycle traits (e.g., resting cyst formation, zygote formation, and vegetative growth). Further insights into the dynamics of the relationship between bloom-forming species and climate change will support the development of small-scale predictive models that can assist in reducing and/or mitigating the severe impacts of HABs worldwide.

## Figures and Tables

**Figure 1 microorganisms-11-01874-f001:**
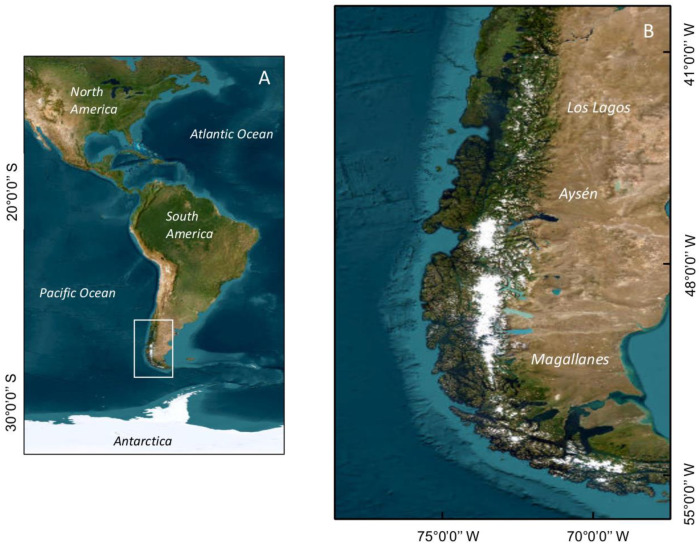
Map showing (**A**) South America; (**B**) Chilean Patagonia, where the impacts of HAB events have been most severe. The administrative regions (Los Lagos, Aysén, and Magallanes) are indicated.

**Figure 2 microorganisms-11-01874-f002:**
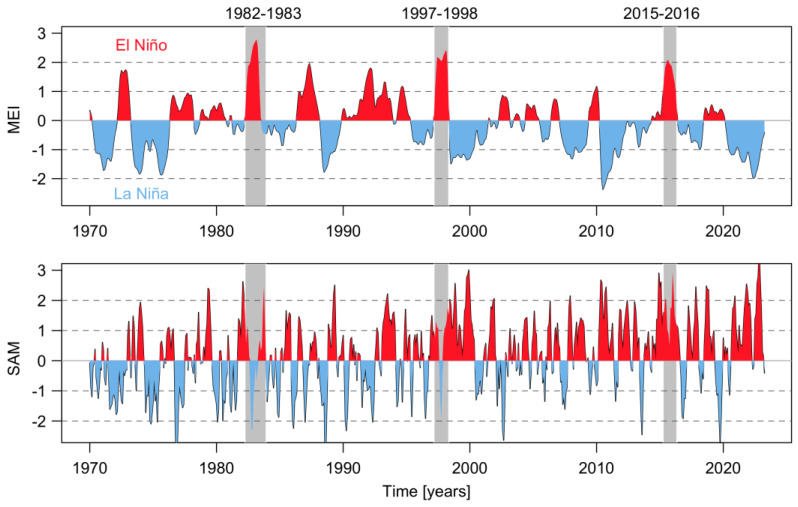
Temporal evolution of the 3-month running mean of the Multivariate ENSO Index (MEI) version 2 (MEI.v2) and the Southern Annular Mode (SAM) from January 1970 to April 2023. The gray rectangle shows the more intense ENSO in a positive phase.

**Figure 3 microorganisms-11-01874-f003:**
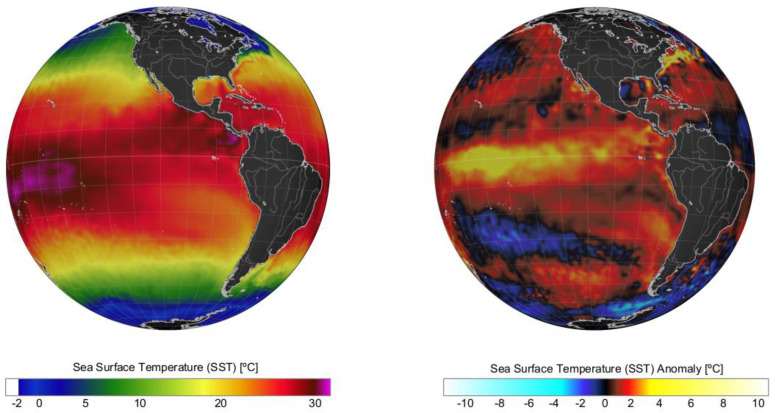
Sea surface temperature (SST, **left** panel) and SST anomalies (**right** panel) recorded on a global scale in January 2016 during an intense El Niño. Data obtained from the US National Weather Service.

**Figure 4 microorganisms-11-01874-f004:**
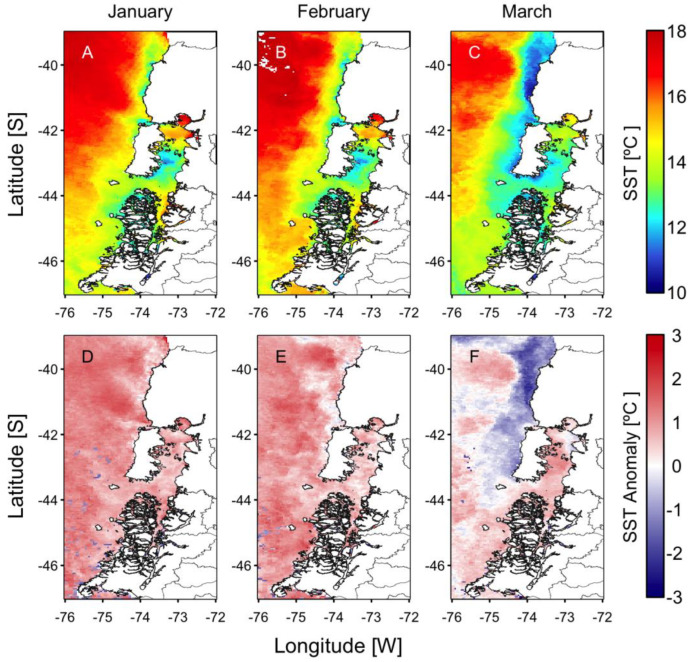
Monthly sea surface temperature (SST, (**A**–**C**) upper panels) and SST anomalies ((**D**–**F**) lower panels) from January to March. Anomalies were computed with the 2003–2015 monthly means from MODIS images.

**Figure 5 microorganisms-11-01874-f005:**
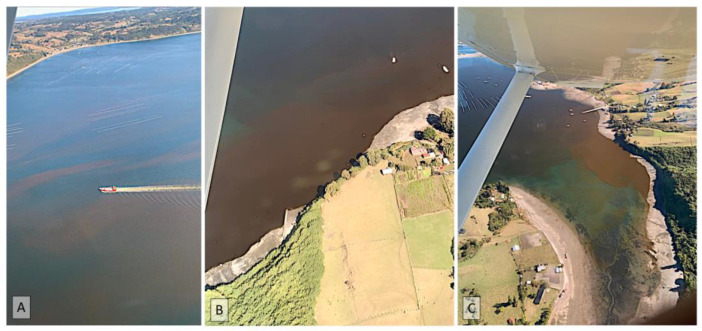
(**A**–**C**) Red tide caused by *Heterosigma akashiwo*. The aerial photos were taken over the Chiloé Inland Sea in March 2022.

**Figure 6 microorganisms-11-01874-f006:**
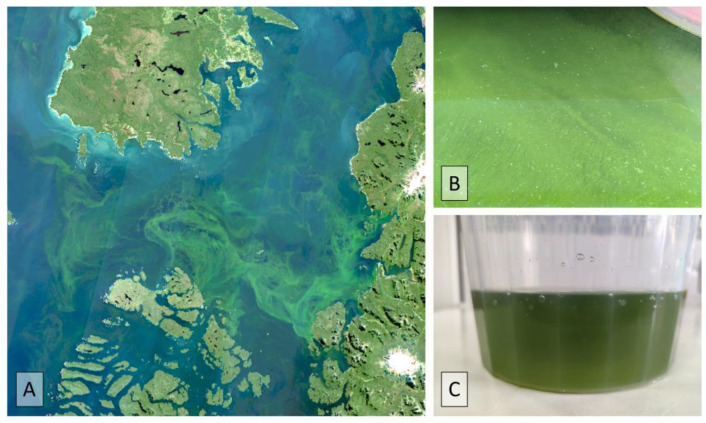
(**A**) True-color Sentinel-2 satellite image of green filaments observed in Corcovado Gulf, Southern Chile; (**B**) green filaments of *L. chlorophorum*, and (**C**) water sample from Tenglo Channel, city of Puerto Montt.

**Figure 7 microorganisms-11-01874-f007:**
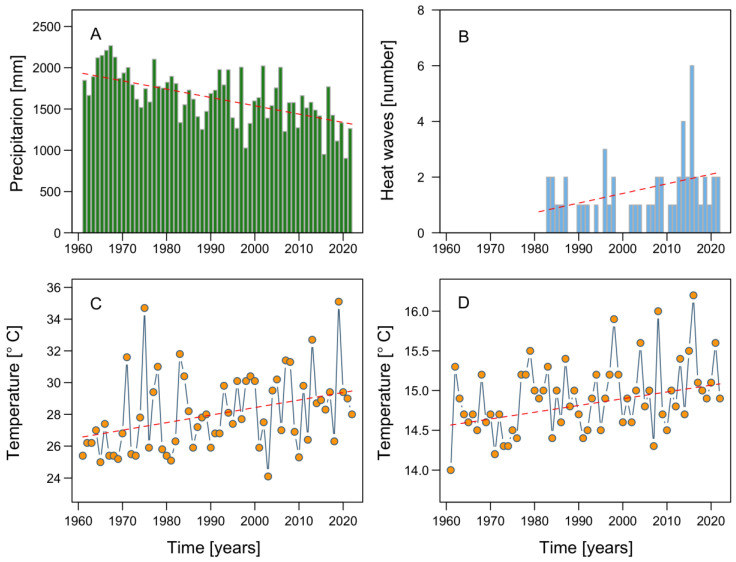
Climatological parameters recorded at the meteorological station located at Tepual airport, Puerto Montt, from 1961 to 2022: (**A**) total precipitation; (**B**) number of heat waves; (**C**) maximum air temperature; (**D**) mean temperature.

**Figure 8 microorganisms-11-01874-f008:**
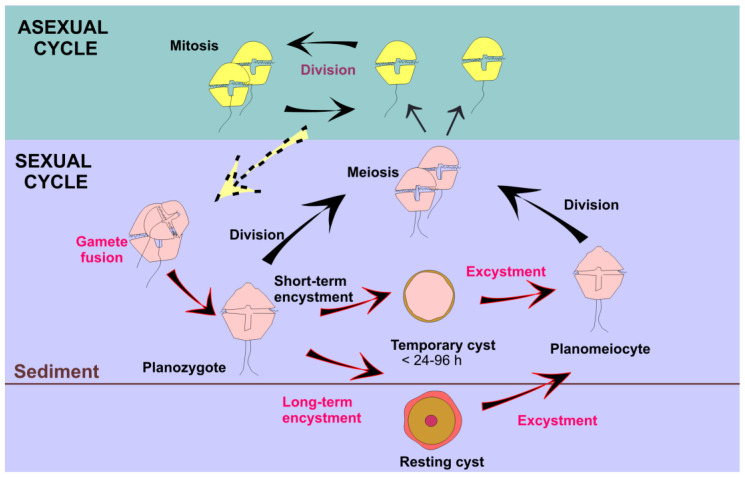
Typical dinoflagellate life cycle. In red, processes that may be affected by temperature changes.

## Data Availability

Not applicable.

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
