# Peer review of "Toxic Algal Bloom Recurrence in the Era of Global Change: Lessons from the Chilean Patagonian Fjords"

_microorganisms, 2023, doi:10.3390/microorganisms11081874_

Round 1
Reviewer 1 Report
This is a very meaningful article that provides us with a lot of information on harmful algal blooms and their impact on fisheries, which is very valuable. It is recommended to have specialized channels that allow scholars from around the world to provide this information and work together to tackle the formation mechanisms of harmful algal blooms. The information provided by the author on typical harmful algal blooms is very scarce and important. Although much of the article is descriptive and lacks in-depth event analysis and cause mining, I still think it is worth publishing. I suggest that the author conduct more analysis of the causes of HABs, in order to provide reference for more harmful red tide researchers.
1. Line 11, there is an extra space in the word 'HABs', or is this a new abbreviation proposed by the author?
2. Line 11, 'HABs' is an abbreviation for “Harmful algal blooms”, not 'Harmful and toxic algal blooms'.
3. Figures, the resolution of Figure 1, 2, 4, 5 and 7 are low, please use the high-definition version.
4. Climate modulation: from global to local responses, this paragraph provides a good analysis of the climate background and typical events, but lacks correlation analysis with harmful algal blooms.
5. Line 148, HABs
6. Extreme HAB events in Chilean Patagonia, this paragraph and subsequent paragraphs are mostly descriptions without in-depth analysis of the reasons.
Author Response
Reviewer #1
Comments and Suggestions for Authors
This is a very meaningful article that provides us with a lot of information on harmful algal blooms and their impact on fisheries, which is very valuable. It is recommended to have specialized channels that allow scholars from around the world to provide this information and work together to tackle the formation mechanisms of harmful algal blooms. The information provided by the author on typical harmful algal blooms is very scarce and important. Although much of the article is descriptive and lacks in-depth event analysis and cause mining, I still think it is worth publishing. I suggest that the author conduct more analysis of the causes of HABs, in order to provide reference for more harmful red tide researchers.
A: Thank you very much for your valuable comments. Additional information on specific aspects of the events analyzed in the manuscript was incorporated.
1. Line 11, there is an extra space in the word 'HABs', or is this a new abbreviation proposed by the author?
A: This was a mistake. Corrected.
2. Line 11, 'HABs' is an abbreviation for “Harmful algal blooms”, not 'Harmful and toxic algal blooms'.
A: Corrected.
3. Figures, the resolution of Figure 1, 2, 4, 5 and 7 are low, please use the high-definition version.
A: We appreciate your suggestion, but we consider the figures to be of very good quality.
4. Climate modulation: from global to local responses, this paragraph provides a good analysis of the climate background and typical events, but lacks correlation analysis with harmful algal blooms.
A: Additional information on the impact of the different phases of the long-term climatic cycles used (ENSO, SAM and PDO) were incorporated to better connect the information with the occurrence and intensity of HABs.
5. Line 148, HABs
A: Corrected.
6. Extreme HAB events in Chilean Patagonia, this paragraph and subsequent paragraphs are mostly descriptions without in-depth analysis of the reasons.
A: Additional information on specific aspects of the events analyzed (e.x. cell densities reached) in the manuscript were incorporated.

Reviewer 2 Report
The research needs careful language review, as the sentences are tall and untidy.
Most of the research is structural and often needs to describe the events of algal blooming.
The authors should have mentioned some about the concentrations of nutrient salts, which are an essential factor for the blooming of algae.
As the authors did not mention the cell numbers of each bloom, they also did not mention the species that bloom and cause harm to the fish, but they did not secrete toxins.
Abstract:
The abstract is just an introduction to the review and does not include any summaries or results. For this, I want to rephrase it in a good way so that the time period covered in the review and the most important events that appeared in the research area are mentioned, as well as the factors that are likely to be the cause of the phenomenon, as well as the species that were the cause of this phenomenon.
Introduction:
A detailed description of the study area must be provided.
A brief definition of the El Niño Southern Oscillation (ENSO), the Southern Annular Mode (SAM) and the Pacific Decadal Oscillation must be made so that the reader understands its meaning.
Climate Modulation: from global to local responses:
Could you please clarify what you mean by “climate modulation: from global to local responses”?
Line 126-131: you mentioned (The combined reduction in freshwater inputs into the coastal system weakened the superficial haline stratification, allowing the entry of deeper water of higher salinity and rich in nutrients into superficial layers of the inland sea of NW Patagonia, exacerbated by the continuous upwelling pulses of oceanic water masses that occurred at the adjacent continental shelf. A laboratory simulation of these high salinity and temperature conditions showed that they are optimal for the growth of P. verruculosa [13,35]).
The results showed that the optimal growth conditions for H. akashiwo and P. verruculosa differed, with a maximum growth of H. akashiwo obtained at 15 °C and a salinity of 20 psu, and 18 °C and a salinity of 30 psu for P. verruculosa as mentioned by Sanhueza et al (13). Please Rephrase the sentence correctly and accurately.
Extreme HAB events in Chilean Patagonia:
Lines 137-142: This sentence was written several times, so I want to delete this part and enter it directly into the topic.
Lines 147-152: You said (In the following, several high-intensity HAB events in Chilean Patagonia in recent decades are discussed, including those responsible for biotoxin-mediated fish kills and those resulting in high-biomass microalgal accumulations. The relationship of these events to environmental and anthropogenic factors and the species-specific responses to them are considered as well). Please mention the recent events with the date and the species and causes of the event, the number of species and the damages that caused this event.
HABs of biotoxin-producing microalgae:
Lines 165-182, you mentioned some harmful algal blooms; therefore, it is necessary to mention the species, the conditions surrounding the phenomenon, and the concentrations of nutrient salts.
Fish-killing microalgal blooms
Lines 211-214, The overwhelming of P. verruculosa, H. akashiwo, and Karenia spp. in Chilean salmon farms causes massive mortalities [10,12,31]; that industry is considered the world's second-largest farmed salmon producer [51].
Future perspectives: potential species responses, trends, and unresolved topics
Lines 404- 423 This paragraph is unnecessary; it is just a narration from the books.
Author Response
Reviewer #2
Comments and Suggestions for Authors
The research needs careful language review, as the sentences are tall and untidy. Most of the research is structural and often needs to describe the events of algal blooming. The authors should have mentioned some about the concentrations of nutrient salts, which are an essential factor for the blooming of algae. As the authors did not mention the cell numbers of each bloom, they also did not mention the species that bloom and cause harm to the fish, but they did not secrete toxins. Without such an analysis, it is very hard to see the effect that all of these factors had (or did not have) on production.
A: This manuscript has been edited and revised by a native English editor. However, we have tried to shorten and clarified the text following the reviewer comment. Regarding the data presented, we have here reviewed the available data on the blooms described, adding now data on cell numbers as the reviewer has suggested. We have also added some explanation about the fish-killing and non-toxic microalgal blooms.
Abstract:
The abstract is just an introduction to the review and does not include any summaries or results. For this, I want to rephrase it in a good way so that the time period covered in the review and the most important events that appeared in the research area are mentioned, as well as the factors that are likely to be the cause of the phenomenon, as well as the species that were the cause of this phenomenon.
A: The abstract has been modified following the reviewer comments.
Introduction:
A detailed description of the study area must be provided.
A: A detailed paragraph with the description of the study area was incorporated in the introduction.
A brief definition of the El Niño Southern Oscillation (ENSO), the Southern Annular Mode (SAM) and the Pacific Decadal Oscillation must be made so that the reader understands its meaning.
A: A description of each large-scale climate cycles (ENSO, PDO and SAM) was included.
Climate Modulation: from global to local responses:
Could you please clarify what you mean by “climate modulation: from global to local responses”?
A: This heading has been clarified and changed to: “Climate modulation: a global phenomenon affecting microalgal local responses”
Line 126-131: you mentioned (The combined reduction in freshwater inputs into the coastal system weakened the superficial haline stratification, allowing the entry of deeper water of higher salinity and rich in nutrients into superficial layers of the inland sea of NW Patagonia, exacerbated by the continuous upwelling pulses of oceanic water masses that occurred at the adjacent continental shelf. A laboratory simulation of these high salinity and temperature conditions showed that they are optimal for the growth of P. verruculosa [13,35]). The results showed that the optimal growth conditions for H. akashiwo and P. verruculosa differed, with a maximum growth of H. akashiwo obtained at 15°C and a salinity of 20 psu, and 18°C and a salinity of 30 psu for P. verruculosa as mentioned by Sanhueza et al. [13]. Please rephrase the sentence correctly and accurately.
A: The reviewer is right, but the differences between both species are more complex because H. akashiwo growths over a wider range of temperatura and salinity as we explained later in the manuscript. As suggested by the reviewer, we have added in this paragraph the optimal growth conditions of temperatura and salinity for P. verruculosa.
Extreme HAB events in Chilean Patagonia:
Lines 137-142: This sentence was written several times, so I want to delete this part and enter it directly into the topic.
A: The paragraph has been deleted to enter directly into the topic.
Lines 147-152: You said (In the following, several high-intensity HAB events in Chilean Patagonia in recent decades are discussed, including those responsible for biotoxin-mediated fish kills and those resulting in high-biomass microalgal accumulations. The relationship of these events to environmental and anthropogenic factors and the species-specific responses to them are considered as well). Please mention the recent events with the date and the species and causes of the event, the number of species and the damages that caused this event.
A: This is an introductory paragraph. We have included now in it the name of the species that caused the events, and more information on dates and damages on the following paragraphs, as the reviewer has indicated.
HABs of biotoxin-producing microalgae:
Lines 165-182, you mentioned some harmful algal blooms; therefore, it is necessary to mention the species, the conditions surrounding the phenomenon, and the concentrations of nutrient salts.
A: As explained in the previous response, we have completed this section with more data about the years of the most important blooms and the cell densities reached.
Fish-killing microalgal blooms
Lines 211-214, The overwhelming of P. verruculosa, H. akashiwo, and Karenia spp. in Chilean salmon farms causes massive mortalities [10,12,31]; that industry is considered the world's second-largest farmed salmon producer [51].
A: Effectively, that's right. The paragraph indicated is well written.
Future perspectives: potential species responses, trends, and unresolved topics
Lines 404- 423 This paragraph is unnecessary; it is just a narration from the books.
A: In the same way as a brief definition of ENSO and SAM has been required by the reviewer, it seems necceesary a brief summary of the dinoflagellates life cycle to explain how global change related factors may affect their reproductive strategy. Therefore,we think that this paragraph, written following our own knowledge on the topic, should be mantained.

Reviewer 3 Report
The manuscript describes important and unique information on the local HABs in Chile. I enjoyed reading and learned quite a lot. The following are my questions and suggestions.
Line 34: Please specify "direct connection is only at a local level." I think you meant "direct connection between climate change and HABs." Otherwise, people in the world are directly affected by climate change.
Line 40: Also environment damage too.
Line 48: Are you sure these blooms are caused by climate change? References are needed. The next paragraph states, "The role of climate change in HAB is poorly understood."
Line 205: The four species were equally detected (cell count wise)? Approximately how many?
Line 217: I thought this was an event of P. verruculosa followed by A. catenella, and together lost US800 million salmon. Am I wrong? This sentence states only P. verruculosa was the cause of the damage.
Line 252: Could you describe the average water temperature and salinity of the area? How often does it reach salinity 30 and temperature 18C?
Line 261: I think it is better to define high biomass here. Something like "non-toxin producer but accumulated biomass can be toxic"
Line 269: We don't have much infomation on Lepidodinium. Could you add? Are they chilean origin, or where does it come from? Are you sure this is not a toxin-producer? Last I read about this species was "unknown" toxicity.
Line 312: Could you add an example of eutrophication increased before and after salmon industry was established? You may want to add a statement that salmon was not indigenous to Chile and when the farm started in the area, so the readers understand the eutrophication issue.
Line 332: I understand P. verruculosa caused the world’s worst HAB damage in 2016, but is this species repeatedly causing blooms in Patagonia? I am only aware of the 2016 event. Otherwise, I wonder why it was a one-time event.
Line 332: This paragraph may be combined with line Line 217.
Line 350: Glad to know that there is a protocol now. So, the dead fish by HABs must be transferred by a truck and burnt is the current protocol?
Author Response
Reviewer #3
Comments and Suggestions for Authors
The manuscript describes important and unique information on the local HABs in Chile. I enjoyed reading and learned quite a lot. The following are my questions and suggestions.
A: Thank you very much for your valuable comments.
Line 34: Please specify "direct connection is only at a local level." I think you meant "direct connection between climate change and HABs." Otherwise, people in the world are directly affected by climate change.
A: This phrase has been modified accordingly.
Line 40: Also environment damage too.
A: The sentence has been changed accordingly.
Line 48: Are you sure these blooms are caused by climate change? References are needed. The next paragraph states, "The role of climate change in HAB is poorly understood."
A: Three references that support this paragraph were incorporated. The next paragraph (The role of climate………) was also corrected.
Díaz, P.A., Pérez-Santos, I., Basti, L., Garreaud, R., Pinilla, E., Barrera, F., Tello, A., Schwerter, C., Arenas-Uribe, S., Soto-Riquelme, C., Navarro, P., Díaz, M., Álvarez, G., Linford, P., Altamirano, R., Mancilla-Gutiérrez, G., Rodríguez-Villegas, C., Figueroa, R.I., 2023. How local and climate change drivers shaped the formation, dynamics and potential recurrence of a massive fish-killer microalgal bloom in Patagonian fjord. Sci. Total Environ. 865, 161288.
León-Muñoz, J., Urbina, M.A., Garreaud, R., Iriarte, J.L., 2018. Hydroclimatic conditions trigger record harmful algal bloom in western Patagonia (summer 2016). Sci. Rep. 8, 1330.
Trainer, V.L., Moore, S.K., Hallegraeff, G., Kudela, R.M., Clement, A., Mardones, J.I., Cochlan, W.P., 2020. Pelagic harmful algal blooms and climate change: Lessons from nature’s experiments with extremes. Harmful Algae 91, 101591.
Line 205: The four species were equally detected (cell count wise)? Approximately how many?
A: The maximum cell densities for each of the four species were incorporated.
Line 217: I thought this was an event of P. verruculosa followed by A. catenella, and together lost US800 million salmon. Am I wrong? This sentence states only P. verruculosa was the cause of the damage.
A: That's a very accurate question from the reviewer. The losses (US800 million dollars) in the salmon industry were only attributed to the bloom of P. verruculosa. The subsequent bloom of A. catenella did not affect salmon farming. In this case, those affected were bivalve producers (mainly mussels Mytilus chilensis) and artisanal fishing. Unfortunately, these losses are not quantified.
Line 252: Could you describe the average water temperature and salinity of the area? How often does it reach salinity 30 and temperature 18C?
A: The following paragraph was included in the text: These environmental conditions are not typical of the fjord system, especially in areas with high contributions of fresh water from mighty rivers where there is strong haline strat-ification in the first 10 m, such as the Reloncaví Fjord in Los Lagos region, but they can occur during periods of extreme hydroclimatic anomalies such as those that occurred in the summer of 2016 [46].
Line 261: I think it is better to define high biomass here. Something like "non-toxin producer but accumulated biomass can be toxic"
A: Thanks for the suggestion, a definition of “high biomass species” has been added.
Line 269: We don't have much infomation on Lepidodinium. Could you add? Are they chilean origin, or where does it come from? Are you sure this is not a toxin-producer? Last I read about this species was "unknown" toxicity.
A: Lepidodinium is not known to produce toxicity humans or marine organisms, although its, blooms have been associated with mass mortalities of fishes and cultured bivalves, which has been related to the consequences of high biomass side effects and the secretion of extracellular polymeric substances (Roux et al. 2021, Scientific Reports).
Line 312: Could you add an example of eutrophication increased before and after salmon industry was established? You may want to add a statement that salmon was not indigenous to Chile and when the farm started in the area, so the readers understand the eutrophication issue.
A: We have added a sentence to clarify this aspect.
Line 332: I understand P. verruculosa caused the world’s worst HAB damage in 2016, but is this species repeatedly causing blooms in Patagonia? I am only aware of the 2016 event. Otherwise, I wonder why it was a one-time event.
A: Blooms of P. verruculosa have previously occurred in Patagonia, but with minor impacts. Mardones et al. (2012) documented the presence of the species for 7 years between 2004 and 2012, although the densities never exceeded 400 cells mL-1. For this reason, the 2016 event is unique in terms of cell density reached (23,000 cells mL-1) and socioeconomic impacts.
Mardones, J.I., Clément, A., Rojas, X., 2012. Monitoring potentially ichthyotoxic phytoflagellates in southern fjords of Chile. Harmful Algae News 45, 6-7.
Line 332: This paragraph may be combined with line Line 217.
A: We appreciate your suggestion. However, the second paragraph is in the context of the massive dumping of dead salmon that took place in the Pacific Ocean as a "mitigation" measure. We prefer to keep the paragraphs separate.
Line 350: Glad to know that there is a protocol now. So, the dead fish by HABs must be transferred by a truck and burnt is the current protocol?
A: After the 2016 event, the storage capacities for dead fish were significantly increased in farming centers and optimal fish processing plants for fishmeal. In addition, the season of greatest risk of HAB occurrences, all companies must hire large-capacity vessels (deep-sea fishing vessels) that quickly extract and transport mortality. Despite this, during events like the one in 2016, these types of protocols will always be put to the test.
The next paragraph was incorporated into the text to clarify the idea: The new measures included an increase in the storage capacity of dead fish in the farming centers and greater transport and processing capacity for this biomass. In addition, some mitigation systems have been implemented, such as the use of bubble curtains [89].

Round 2
Reviewer 2 Report
I agree with the corrections while adhering to the corrections written in the research itself.